# Community ageing research 75+ study (CARE75+): an experimental ageing and frailty research cohort

Anne Heaven,[1] Lesley Brown,[1] John Young,[2] Elizabeth Teale,[2] Rebecca Hawkins,[2] Karen Spilsbury,[3] Gail Mountain,[4] Tracey Young,[5] Victoria Goodwin,[6] Barbara Hanratty,[7] Carolyn Chew-Graham,[8] Caroline Brundle,[1] Farhat Mahmood,[1] Ikhlaq Jacob,[1] Amrit Daffu-O'Reilly,[9] Andrew Clegg[2]

For numbered affiliations see end of article.

**Correspondence to**
Dr Andrew Clegg;
a.p.clegg@leeds.ac.uk

## ABSTRACT

**Introduction** The Community Ageing Research 75+ Study (CARE75+) is a longitudinal cohort study collecting an extensive range of health, social and economic data, with a focus on frailty, independence and quality of life in older age. CARE75+ is the first international experimental frailty research cohort designed using Trial within Cohorts (TwiCs) methodology, to align applied epidemiological research with clinical trial evaluation of interventions to improve the health and well-being of older people living with frailty.

**Methods and analysis** Prospective cohort study using a TwiCs design. One thousand community-dwelling older people (≥75 years) will be recruited from UK general practices. Nursing home residents, those with an estimated life expectancy of 3 months or less and people receiving palliative care will be excluded. Data collection assessments will be face to face in the person's home at baseline, 6 months, 12 months, 24 months and 48 months, including assessments of frailty, cognition, mood, health-related quality of life, comorbidity, medications, resilience, loneliness, pain and self-efficacy. A modified protocol for follow-up by telephone or web based will be offered at 6 months. Consent will be sought for data linkage and invitations to additional studies, including intervention studies using the TwiCs design. A blood sample biobank will be established for future basic science studies.

**Ethics and dissemination** CARE75+ was approved by the NRES Committee Yorkshire and the Humber—Bradford Leeds 10 October 2014 (14/YH/1120). Formal written consent is sought if an individual is willing to participate and has capacity to provide informed consent. Consultee assent is sought if an individual lacks capacity. Study results will be disseminated in peer-reviewed scientific journals and scientific conferences. Key study results will be summarised and disseminated to all study participants via newsletters, local older people's publications and local engagement events. Results will be reported on a bespoke CARE75+ website.

**Trial registration number** ISRCTN16588124;Results stage

## Strengths and limitations of this study

► Community Ageing Research 75+ Study (CARE75+) is a prospective cohort study recruiting older people aged 75 and over, designed using Trial within Cohorts (TwiCs) methods, collecting an extensive range of demographic, health and socio-economic data at baseline, 6, 12, 24 and 48 months.

► Our recruitment strategy, including home consent visits, home assessments and use of researchers with community language skills, is designed to optimise the recruitment of older people across the frailty spectrum.

► CARE75+ will recruit participants from a variety of ethnic backgrounds and those with advanced frailty who are often under-represented in research.

► Care home residents are not eligible for the study, aligned with the TwiCs design, meaning that findings cannot be generalised to this group of especially frail older people.

► CARE75+ is a cohort of high strategic relevance, which will help shape future UK and international health and research policy in ageing and frailty.

## INTRODUCTION

Global ageing demographic projections indicate that there will be two billion people aged over 65 worldwide by 2050.[1][2] Frailty is an especially problematic expression of population ageing, with profound implications for planning and delivery of health and social care services globally. It is a condition characterised by loss of biological reserves, failure of homoeostatic mechanisms and increased vulnerability to adverse outcomes following relatively minor stressor events.[3][4] Thus, a mild infection, new medication or minor surgery can result in a sudden, disproportionate change in health status or functional status for an older person with frailty, for example, a change from independence to dependence, a fall, or development of delirium. Frailty is also associated with an increased risk of a range of adverse outcomes, including future disability, admission to hospital, long-term care residence and mortality.[5]

To date, the healthcare response to frailty has been predominantly reactive and secondary care based. However, there is increasing recognition that frailty should be identified and managed as a long-term condition with preventative and proactive care models.[6–8] Furthermore, with the widespread introduction of robustly developed tools to detect frailty in primary care such as the electronic Frailty Index (eFI) based on routinely available primary care electronic health record (EHR) data in the UK,[9] primary care teams can now more readily and reliably identify older people with frailty within their patient populations. These novel approaches are providing opportunities to develop and deliver services according to frailty status rather than chronological age.

Improved management of frailty requires an integrated approach spanning primary care, secondary care and social services that incorporates consideration of frailty transitions and health trajectories. Where possible, integrated care pathways should be developed and implemented based on suitably targeted, evidence-based interventions. Although recruitment to ageing and frailty observational research studies has historically been relatively high,[10 11] recruitment rates to clinical trials of frailty interventions have frequently been low.

The Trial within Cohorts (TwiCs) design[12] is an innovative research methodology that has the potential to enhance participation of older people with frailty in a range of studies including clinical trials, and to increase the capacity to conduct high quality frailty research.[13] The TwiCs design has several key features including the establishment of an observational cohort to both provide longitudinal data and function as a recruitment platform for multiple trials and other research studies. Each individual trial uses random selection of some (not all) participants from the cohort; intervention-centred information and consent is applied. The process aims to replicate the real world of routine healthcare by taking informed consent only from those randomised to receive an intervention, as the ongoing cohort study provides a natural control group.

## METHODS AND ANALYSIS
### Aim
Our aim is to establish a longitudinal cohort of older people to investigate frailty, disability and quality of life in older age and to act a recruitment platform for future studies (substudies) to enable the development and evaluation of interventions to improve outcomes for older people.

### Patient and public involvement
We have established a Frailty Oversight Group (FOG) as a central component of the Community Ageing Research 75+ Study (CARE75+) study. The FOG comprises a core reference group of four key individuals with links to local community organisations involved in the support of older people living with frailty, and a minority ethnic group advocate from the local authority. The FOG plays a key role in developing research questions for the cohort, including reviewing any proposed data analyses or nested studies.

The FOG had close involvement in developing and piloting the outcome assessment schedule for the study, highlighting the need to include measures that extend beyond traditional health domains into areas such as loneliness and resilience in later life. The FOG contributed to the development of all study materials, including invitation letters and participant information sheets, to ensure alignment with the needs of older people. Results are disseminated widely to participants, including through regular newsletters and an annual celebration event.

### Design
A multisite, community-based cohort study using a TwiCs design.[12]

### Inclusion criteria
Community-dwelling older people aged ≥75 years.

### Exclusion criteria
People with terminal cancer, life expectancy of 3 months or less and people in receipt of palliative care services will be excluded. Care home residents and people living at home who are bedbound will be excluded. However, we will attempt to follow up people who transition to a care home during the course of the study.

### Assessments
The CARE75+ assessment includes detailed information on the demographic, health and social circumstances of participants. An extensive range of measures is collected using validated instruments, including assessments of frailty, cognition, mood, health-related quality of life, comorbidity, medications, resilience, loneliness and self-efficacy (table 1). The selected measures have been carefully chosen to ensure that CARE75+ includes measures with the necessary validity, reliability and responsiveness to enable both applied epidemiological investigation and randomised trial evaluation of future interventions to improve outcomes.

### List of current assessments
► Demographic information (age, sex, ethnicity, marital status, living circumstances, housing type, education, previous occupation).
► Family networks and informal support (self-report).
► Resource use: general practitioner (GP), hospital and outpatient admissions. Use of aids and adaptations (self-report).
► Formal care (self-report).
► Smoking habits and alcohol consumption (self-report).
► Vision LogMar Vision test[14] (Thompson Software Solutions).[15]
► Hearing (the Whispered Voice test).[16]
► Sleep (self-report).

**Table 1** Domains and associated measures included in Community Ageing Research 75+ Study assessment schedule

| Domain | Measures |
| --- | --- |
| Sociodemographic | Age<br>Gender<br>Housing type<br>Room temperature<br>Education<br>Occupation<br>Qualifications<br>Family information<br>Formal support<br>Informal support<br>Smoking<br>Alcohol |
| Anthropometrics | Height<br>Weight<br>Body mass index<br>Bioelectric impedance analysis |
| General health data | Blood pressure<br>Hearing<br>Vision<br>Comorbidities<br>Medications<br>Falls<br>Sleep |
| Frailty | English Longitudinal Study of Ageing Frailty Index (FI)<br>Electronic FI<br>Phenotype model<br>Clinical Frailty Scale, 7-category version<br>Edmonton Frail Scale |
| Health-related quality of life | Short-Form 36 Item Health Questionnaire<br>EuroQol 5-Dimension Health Questionnaire, five-level version |
| Cognition | Montreal Cognitive Assessment |
| Activities of daily living (ADL) | Barthel Index<br>Nottingham Extended ADL |
| Mobility | Timed-up-and-go test<br>Gait speed<br>Walking aid |
| Muscle strength | Grip strength |
| Pain | Geriatric pain measure |
| Loneliness | De Jong Gierveld Loneliness Scale |
| Depression | Geriatric Depression Scale |
| Resilience | Brief Resilience Scale |
| Self-efficacy | General Self-Efficacy Scale |

► Medication (prescribed) details (name, dose, frequency) will be collected from primary care EHR. Non-prescribed medication will be self-reported.

► Cognitive function assessed using the Montreal Cognitive Assessment (MoCA),[17] a brief cognitive assessment instrument. The MoCA assesses different cognitive domains: attention and concentration; executive function; memory; language; conceptual thinking; calculations and orientation. The total possible score is 30, with higher scores indicating better cognitive function, and a score of ≥26 considered normal.

► Comorbidities data, collected via the primary care EHR and by self-report using the Katz comorbidity questionnaire.[18] This questionnaire asks questions on various health conditions requiring a 'yes' or 'no' response.

► General health and health-related quality of life, using the RAND Short-Form 36-Item Health Survey (SF-36)[19] which includes 36 questions spanning eight health domains: physical functioning; bodily pain; role limitations due to physical health problems; role limitations due to personal or emotional problems; general mental health; social functioning; energy/fatigue and general health perceptions. It also includes a single item that provides an indication of perceived change in health. The SF-36 enables calculation of Physical Component Summary and Mental Component Summary scores, and derivation of an overall Health Utility Score, the Short-Form Six Dimension score suitable for use in economic evaluations.[20]

► Health-related quality of life using the EuroQol Five Dimension Health Questionnaire (five-level version) EQ-5D-5L.[21] The EQ-5D-5L five dimensions are: mobility, self-care, usual activities, pain/discomfort and anxiety/depression. Each dimension has five levels of severity: no problems, slight problems, moderate problems, severe problems and extreme problems. The scores for each of the five dimensions are combined in a five-digit number representing health status that can be converted into a utility index (0 for dead, 1 for perfect health and negative values for states worse than death) for use in economic evaluations.

► Basic activities of daily living (ADL) using the Barthel Index (BI).[22] The BI assesses functional status on a 20-point scale by recording ability to complete ten basic ADL; bathing, bladder function, bowel function, dressing, feeding, grooming, mobility, stairs, toilet use and transfers. Higher scores indicate greater independence.

► Instrumental ADL, measured using the Nottingham Extended ADL (NEADL) scale.[23] The NEADL includes questions on everyday activities in the domains of mobility, kitchen, domestic and leisure and is scored between 0 and 66, with higher scores indicating greater independence.

► Measures of frailty:
  – Research standard 60-item Frailty Index, based on the cumulative deficit model of frailty,[24] and previously validated as part of the English Longitudinal Study of Ageing.[25] The Frailty Index score is calculated an equally weighted proportion of the

number of deficits present in an individual relative to the total possible.

- The phenotype model of frailty, based on the five physical characteristics as reported in the original Cardiovascular Health Study (slow walking speed, weight loss, exhaustion, weak grip strength, low energy expenditure).[3] Slow walking speed is assessed by a timed 3 m walk and results stratified by height and gender using values described in the original Cardiovascular Health Study, from which the phenotype model was derived.[3] Weight loss is determined by the following question. 'In the last year, have you lost more than 10 pounds unintentionally?' Exhaustion is identified using the following questions: 'How often in the last week do you feel that everything you did was an effort?' and 'could not get going?'. Responses are: rarely or none of the time (<1 day)=0; some or a little of the time (1–2 days)=1; moderate amount of the time (3–4 days)=2; most of the time=3. If the participant answers '2' or '3' to either question they meet the criterion for exhaustion. Hand grip strength is assessed using a Jamar dynamometer and stratified using criteria from the Cardiovascular Health Study[3] with the mean of three attempts calculated for the dominant and non-dominant hand. Low activity is assessed using data obtained from the Physical Activity domain of the SF-36.[19] Those with no characteristics are identified as fit, one or two characteristics as prefrail and three to five characteristics as frail.
- The seven category Clinical Frailty Scale (CFS),[24] which is a validated measure of frailty based on clinical descriptors and pictographs, designed for specialist and non-specialist use in routine clinical practice. The CFS is an ordinal measure, with scores ranging from 1 (fit) to 7 (severe frailty).
- The Edmonton Frail Scale (EFS),[26] which is a validated frailty measure designed for specialist and non-specialist use that records information on nine frailty domains (cognition, general health, functional independence, social support, medication use, nutrition, mood, continence, functional performance). The EFS is scored out of a total of 17, with higher scores indicating increasing frailty.
- The eFI score,[9] based on the cumulative deficit model of frailty, including 36 variables recorded in the primary care EHR as part of routine care. The eFI score is calculated as an equally weighted proportion of the number of deficits present in an individual relative to the total possible. The eFI enables identification of frailty categories (fit, mild frailty, moderate frailty, severe frailty) and is obtained directly from the primary care EHR.
► Height weight and body composition: researcher assessment using bioimpedance scales (Marsden BFA-220P Body fat analyser). Weight loss is obtained

by self-report at baseline and calculated from previously recorded weight data at follow-up time points.
► Blood pressure (Life source auto inflation blood pressure monitor): sitting (three times), standing (once).
► Mobility, calculated using the timed-up-and-go test (TUGT).[27] The TUGT assesses a person's mobility and requires both static and dynamic balance. It measures the time that a person takes to rise from a chair, walk 3 m, turn around, walk back to the chair and sit down. A person's usual walking aid is used if needed. People completing the test in less than 20 s tend to be independently mobile, able to get in and out of a chair without assistance and climb stairs. People completing the test in 20–29 s demonstrate greater variability in mobility, balance and functional ability. Completion of the TUGT in 30 s or more identifies people likely to require assistance with getting in and out of a chair, climbing stairs and leaving the house.
► Pain, measured using the Geriatric Pain Measure Short Form.[28] This questionnaire includes items of pain intensity (current and last 7 days), and dichotomous items on how pain is impacting on a person's mobility, ability to accomplish tasks and to sleep. Items are combined to derive an overall summary score.
► Loneliness recorded using the 11-item De Jong Gierveld Loneliness scale.[29] Subcategories of social and emotional loneliness are calculated and a total score is derived enabling identification of categories: not lonely; moderately lonely; severely lonely; very severely lonely.
► Resilience, measured using the Brief Resilience Scale (BRS).[30] The six items in the BRS include five response options, enabling calculation of an overall score ranging from 1 to 6, with higher scores indicating greater resilience.
► Self-efficacy, measured using the General Self-Efficacy Scale.[31] This scale lists 10 items with 4 response options enabling generation of a summary score ranging from 10 to 40, with higher scores indicating greater resilience.
► Low mood, assessed using the Geriatric Depression Scale Short-Form with a score of ≥5 indicating an abnormal low mood state.[32]
► Self-reported falls.
► Full blood count (Leeds and Bradford sties only): haemoglobin and mean cell volume; red cell (RCC) count; mean cell haemoglobin concentration; mean cell haemoglobin; RCC distribution width, white cell count (including neutrophils, lymphocytes; monocytes; eosinophils; basophils) and platelets.
► Frozen blood aliquots (Leeds and Bradford sites only) for future biochemical analysis, including:
  - Routine biochemistry and haematology: renal profile; liver profile; serum albumin; bone profile; glucose; glycosylated haemoglobin; lipid profile; uric acid; clotting.
  - Endocrine function: cortisol; thyroid function; Insulin-like Growth Factor-1 (IGF-1);

Dehydroepiandrosterone (DHEAS); testosterone; oestradiol; vitamin D; Parathyroid Hormone (PTH); neuronal-specific protein.

- – Immune function: highly sensitive C reactive protein; inflammatory cytokines; rheumatoid factor; markers of immunosenescence.
- – Nutritional markers: vitamin A; vitamins $B_2$, $B_6$, $B_{12}$; vitamin C; ferritin; folate; homocysteine.
- – Biomarkers of ageing: DNA repair capacity; telomere length; markers of oxidative stress.
- – Genetic markers: DNA; RNA; plasma.

The CARE75+ data dictionary is available as an appendix file (see online supplementary additional file 1).

## Assessment schedule

Participants will be assessed at baseline, 6, 12, 24 and 48 months. Face-to-face assessments will be conducted in the participant's home. The feasibility of a modified, telephone-based or web-based assessment protocol will be tested at the 6-month time point for participants who are willing and able to undertake assessments in the alternative formats.

The assessment schedule for CARE75+ (baseline, 6, 12, 24 and 48 months) has been carefully designed to accelerate the frailty translational research pathway by aligning robust epidemiological investigation with the typical follow-up schedule for feasibility and definitive trials of interventions.

## Sample size

The CARE75+ study will generate a comprehensive dataset for applied epidemiological research and will act as a recruitment platform for additional studies (substudies), including qualitative studies as well as randomised controlled trials (RCTs) using TwiCs methods. Therefore, the initial recruitment target is based on appropriate sample size calculations for pilot RCTs of interventions to inform the design of future definitive RCTs alongside applied epidemiological investigation of modifiable component of frailty.

Previous observational studies involving older people with frailty have identified that between 600 and 1000 participants are required for reliable estimates of the main effects.[33] Following an initial pilot phase involving 200 participants to test recruitment methods and gather data on rates of assent to participation in future trials, we plan to recruit 1000 participants over a 4-year period. Previous observational studies involving the oldest old have reported 18-month attrition rates of around 25% due to mortality and withdrawal of consent.[10] As our cohort will include older people with frailty who are at increased risk of adverse outcomes, we plan to recruit a minimum of 250 participants per year thereafter, to maintain a legacy cohort for future clinical trials. Findings from the CARE75+ study will inform the design of a future definitive experimental frailty research cohort of sufficient size to nest a series of definitive intervention trials targeted at a range of potentially modifiable components of frailty, including people living with different frailty severity grades.

## Recruitment

We will work with general practices to identify and recruit participants in primary care. Following initial piloting of recruitment methods in Bradford and Leeds, West Yorkshire, we will extend recruitment to other practices in England, using the skills and experience of staff within the National Institute for Health Research (NIHR) Clinical Research Networks.

The CARE75+ recruitment, consent, assessment and follow-up process is summarised in a study flow chart (figure 1).

## Participant contact

Potential participants will be posted a study invitation pack containing a letter of invitation, a user-friendly participant information leaflet with photographs of the research staff involved in the home visits and a supporting letter from their general practice. Potential participants who are not interested in participating in the study will be invited to contact their general practice to opt out. If potential participants do not opt out, contact details of eligible participants will be provided to the research team via a secure email system. The invitation letter will be followed up after 2 weeks with a telephone call from a researcher to discuss the study in more detail. If initial interest is expressed, the researcher and potential participant will arrange a home visit for an in-depth discussion of the study, where informed, written consent to participate will be sought.

The recruitment methods take into account the range of physical and cognitive challenges encountered by older people. Experience from previous cohort studies involving older people with frailty, disability and cognitive impairment has demonstrated that direct telephone calls or in-person visits are the only reliable methods of finding out whether potential participants are interested in participating, and may be preferred because they are seen as less of a burden.[33] Recruitment procedures will ensure that an older person with frailty receives all the necessary information to make an informed decision about participation. Procedures have been developed in close partnership with lay representatives through our patient and public involvement FOG,[34] established as part of the NIHR Collaboration for Leadership in Applied Health Research and Care, Yorkshire and Humber (NIHR CLAHRC) programme.

## Participant consent

Following initial telephone contact, researchers will visit participants who express an interest in participation and verbally explain the study in detail, including providing a comprehensive study information leaflet. Potential participants will be able to have an advocate, family member or friend present and will be offered 48 hours to reflect on the information before deciding to consent. For

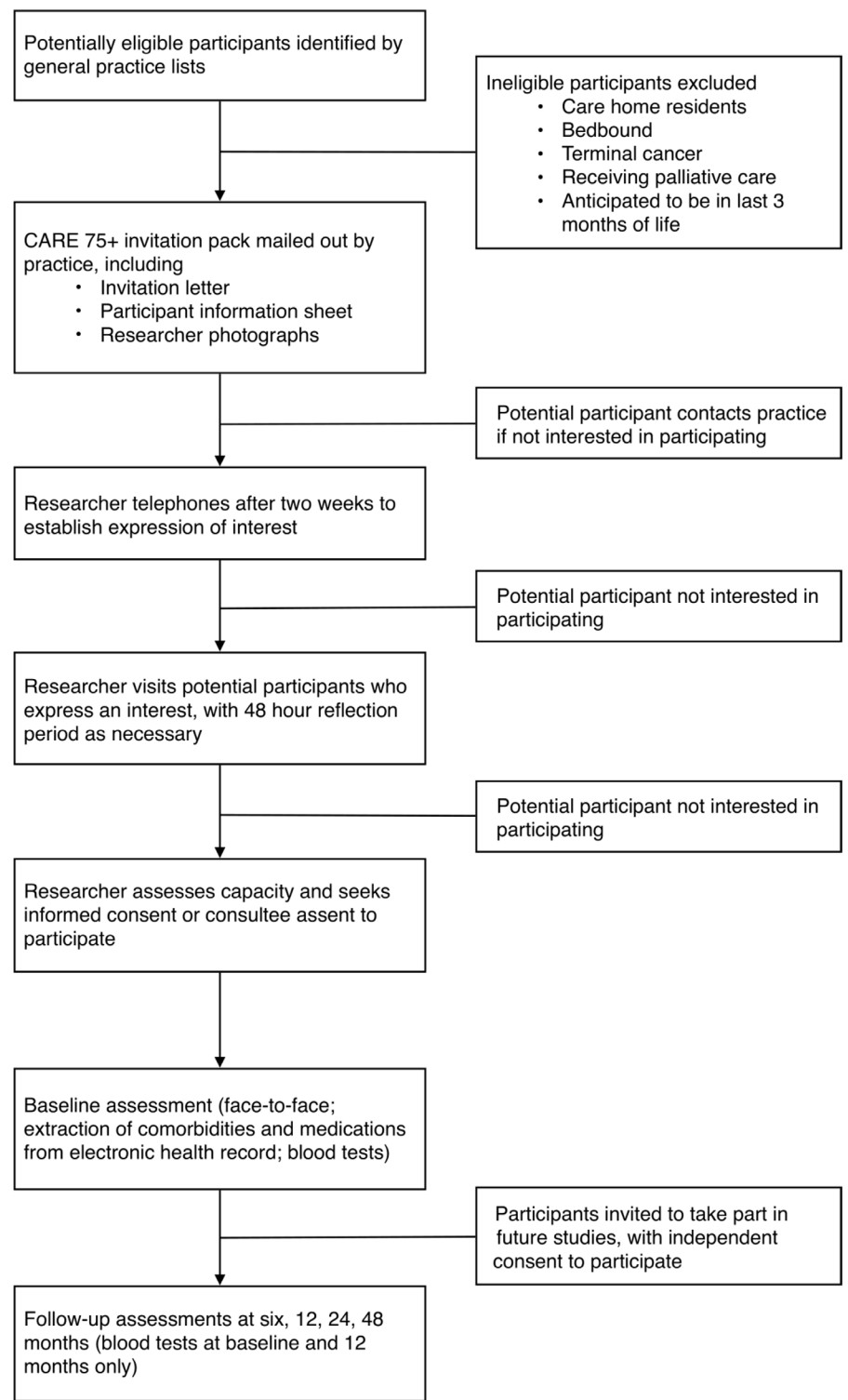

**Figure 1** Data collected during CARE75+ assessments, including all variable and value names and labels. CARE75+, Community Ageing Research 75+ Study.

individuals whose first language is not English, a community language speaking researcher will be assigned where possible or a suitable advocate identified.

Researchers will assess an individual's capacity to consent in accordance with the Mental Capacity Act (MCA).[35] Formal written consent will be sought if an individual is willing to participate and has capacity to provide informed consent. The consent form will detail all processing and disclosure of the information collected including data analysis, data linkage, providing contact details to future researchers, and the storage and use of blood samples. Some components of the consent will be optional (eg, taking and storing blood, consenting to be approached about other studies). Written consultee

assent will be sought if individual participants do not have capacity to consent. Independent consent to participate will be obtained for participation in any future trial.

## Data collection methods

We plan face-to-face data collection, but we will test the feasibility of telephone or web-based modified data collection procedure for participants who are willing, and able, at the 6-month time point.

Prescribed medications, comorbidity data and eFI scores will be obtained from general practice EHRs, extracted using standardised reporting templates developed for the SystmOne[36] and EMISWeb[37] primary care EHR systems.

All data will be collected using a bespoke electronic data capture application (EDCA), the CARE75+ app developed and tested by Tigerteam Software. Blood samples will be collected at baseline and 12 months from participants in the Bradford and Leeds sites.

## Research staff training

Research staff will undertake a bespoke training programme, depending on skills and experience, including: the MCA[35]; research with older people; phlebotomy and safeguarding vulnerable adults. Additionally, staff will receive training in completion of the individual assessment measures and data entry into the EDCA.

## Plans to promote participant retention and complete follow-up

We will seek broad and enduring consent for data linkage and use of collected data following withdrawal or death, aligned with Medical Research Council guidelines for maximising the use of cohort data.[38]

We will post newsletters to participants at least twice a year to provide study updates and encourage continued engagement. We will hold annual engagement events, where feasible to do so, and promote the study locally via affiliated newsletters (eg, Age UK Voice magazine) and local forums.

## Data entry, coding, security and storage

The EDCA will comprise two main components: a Data Collection Application (DCA) and Back Office System (BOS) containing personal identifiable information. The DCA will run on Microsoft Windows platform using an encrypted embedded database to temporarily store data. The BOS database will be on a Microsoft SQL server hosted at Bradford Teaching Hospitals NHS Foundation Trust (BTHFT). All data will be captured off-line in the community. Data will be uploaded regularly to ensure no identifiable data remain on the portable device for longer than 48 hours. Named researchers will have access to the individual details only while data collection takes place. A participant's details will only be released to one researcher at a time via the BOS management system. Access to modules and functions of both the DCA and BOS will be governed by usernames, passwords and role-specific access permissions, to maximise data security.

Remote site data (outside BTHFT) and the on-line completion forms (optional 6-month follow-up protocol) will be transferred to the BIHR-CARE database via the web application auecr.bradfordhospitals.nhs.uk hosted on the web server bhts-bihrweb. The site will be protected by SSL certificates, to encrypt the transfer of data over the internet. Access to the web application https://auecr.bradfordhospitals.nhs.uk on the server bhts-bihrweb will be restricted and protected by the Threat Management Gateway software and SSL certificates. Remote site administrators and researchers will only have access to their own local participants.

Access to the BIHR-CARE database information will be based on role-specific permissions. The chief investigator and project manager will have access to all data, at all levels for administration and governance purposes. Local site administrators will have access to local participant details. Researchers will have access to individual (site-specific) case information only at the time of data collection. Researchers will have a maximum of three participants available on portable devices (laptops) at any one time. Pathology laboratory staff will have access to blood sample data entry pages only. Statisticians and other members of the CARE75+ research team will only have access to pseudo anonymised, that is, those with unique identifiers for use in data linkage or anonymous data. Individual participants will be limited to access to a blank follow-up questionnaire to complete and submit. All submitted data are final and data access is only available to the Super Administrator at BTHFT.

## Data quality

Data quality will be enhanced by integral features of the data capture software, which will identify missing data and outlying values in real time. The software will automatically calculate the total scores for composite assessments. This will increase research efficiency and research data quality by reducing resource required for data cleansing, coding for analysis and reduce inputting errors.

## Statistical methods

We plan interim data analyses after the completion of each stage, that is, baseline, 6, 12, 24 and 48 months follow-up of the study. We will assess frailty transitions using multivariate statistical methods. We will estimate health and social care resource use associated with frailty using economic modelling techniques.

We will conduct applied epidemiological investigation of the association between potentially modifiable components of frailty and outcomes, including: how pain modifies the association between frailty and disability; how resilience modifies the association between frailty and disability; and the association between frailty, mood and outcomes. We will investigate construct and criterion validity of a range of tests collected.[39] We will assess frailty transitions using transition modelling. We will estimate health and social care resource use associated with frailty using economic modelling techniques.

## Methods for any additional analyses (subgroup and adjusted analyses)

Data will be made available to external investigators on request and reviewed by the CARE75+ Data Request Review Committee (DRRC), comprising the chief investigator, CARE75+ project manager, database manager, an independent member and independent lay representative from the FOG.[34]

The ethnic diversity of our planned recruitment sites will enable the investigation of ageing, frailty and disability in different cultural contexts.

## Missing data

Methods for dealing with missing data will depend on the amount of missing data and patterns of missingness for individual variables as part of individual analyses. We will undertake sensitivity analysis to investigate the impact of missing data and we will explore the use of appropriate imputation methods.

## Ethics and dissemination

CARE75+ is an observational study with low risk to participants. Cohort governance will be provided by the National Institute for Health Research Collaboration for Leadership in Applied Health Research and Care Yorkshire and Humber (NIHR CLAHRC YH) Frailty Theme[40] Operational Group comprised the theme leads, theme manager, project managers and coapplicants. Independent scrutiny will be provided by the FOG,[34] which comprised lay members with networks into the wider community of older people in Bradford. Day-to-day monitoring including data quality checks and validations will be the responsibility of a dedicated database manager.

## Access to data

BTHFT will be the data controller for CARE75+. Data will be made available to external researchers in accordance with CARE75+ data sharing protocols following review of the CARE75+ data dictionary (online supplementary file 1) and completion of the CARE75+ data request form (online supplementary file 2), review by the DRRC and completion of a data sharing transfer agreement.

## Ancillary and post-study care

We anticipate that some participants may have potentially unmet care needs and may wish to discuss these with the researcher. We will ensure that researchers are able to signpost participants to local statutory and voluntary organisations (eg, Age UK), or request a GP referral for social services assessment so that appropriate plans can be made for ongoing care.

Safeguarding issues identified during the assessment visits will be reported to the research project manager who will then take advice from the Adult Safeguarding Coordinator in the relevant local authorities.

## Dissemination policy

Study results will be disseminated in peer-reviewed scientific journals and submitted for consideration at local, national and international scientific conferences. Key study results will be summarised and disseminated to all study participants via newsletters, local older people's publication (eg, Voice magazine, Age UK) and local engagement events. Results will be reported on a bespoke CARE75+ website.

Research outputs using data from the CARE75+ study will be required to acknowledge the data source and funder using standardised wording. Additionally, studies involving participants identified from the cohort (substudies) will be required to acknowledge the CARE75+ cohort in all reports. The full protocol and participant level dataset will be made available to not-for-profit investigators. Enquiries should be made to the CARE75+ chief investigator and will be reviewed by the DRRC.

## DISCUSSION

CARE75+ will use novel TwiCs methodology to align applied epidemiological research into ageing and frailty with clinical trials of interventions, potentially accelerating the translational research pathway in this important area.

We describe methods to recruit a cohort of older people and collect an extensive range of health, social and economic outcome data. We plan to collect a range of validated measurements of frailty in CARE75+, including the eFI, which has been made available to every general practice in England through a national implementation project, facilitating the rapid translation of research findings into clinical practice. Our recruitment strategy, including home consent visits, home assessments and use of researchers with community language skills, is designed to optimise the recruitment of older people across the frailty spectrum and from a variety of ethnic backgrounds, including those with advanced frailty who are often under-represented in research. Care home residents are not eligible for the study, aligned with the TwiCs design, meaning that findings cannot be generalised to this group of especially frail older people.

Our vision for CARE75+ is a cohort of high strategic relevance, which will help shape future UK and international health and research policy in ageing and frailty.

**Author affiliations**
[1]Academic Unit of Elderly Care and Rehabilitation, Bradford Institute for Health Research, Bradford, UK
[2]Academic Unit of Elderly Care and Rehabilitation, University of Leeds, Bradford, UK
[3]School of Healthcare, University of Leeds, Leeds, UK
[4]Centre for Applied Dementia Studies, University of Bradford, Bradford, UK
[5]School of Health and Related Research, The University of Sheffield, Sheffield, UK
[6]University of Exeter, PenCLAHRC, Exeter, UK
[7]Institute of Health and Society, Newcastle University, Newcastle upon Tyne, UK
[8]Research Institute for Primary Care and Health Sciences, University of Keele, Keele, UK
[9]Academic Unit of Midwifery, Social Work, Pharmacy and Counselling and Psychotherapy, University of Leeds, Leeds, UK

**Acknowledgements** We are especially grateful to the substantial contributions of members of our Frailty Oversight Group: Anne Grice, Marilyn Foster, Chris McDermott, David Walker, Akhlak Rauf and Ernie Lloyd, who helped to finalise

the protocol, test the assessment procedures and reviewed study materials. We would also like to thank the Newcastle 85+ study team for their invaluable advice and support in setting up CARE75+. Finally, we would like to acknowledge staff from the National Institute for Health Research Clinical Research Networks (NIHR CRNs) in North East and North Cumbria, South West Peninsula, West Midlands, and Yorkshire and the Humber and all participating general practices for their support and enthusiasm in recruiting participants.

**Contributors** AH: substantial contribution to the conception and design of the work; drafting the work and critical revisions; approval of final manuscript; accountable for all aspects of the work. LB: substantial contribution to the conception and design of the work; drafting the work and critical revisions; approval of final manuscript; accountable for all aspects of the work. JY: substantial contribution to the conception and design of the work; drafting the work and critical revisions; approval of final manuscript; accountable for all aspects of the work. ET substantial contribution to the conception and design of the work; drafting the work and critical revisions; approval of final manuscript; accountable for all aspects of the work. RH: substantial contribution to the conception and design of the work; drafting the work and critical revisions; approval of final manuscript; accountable for all aspects of the work. KS: substantial contribution to the conception and design of the work; drafting the work and critical revisions; approval of final manuscript; accountable for all aspects of the work. GM: substantial contribution to the conception and design of the work; drafting the work and critical revisions; approval of final manuscript; accountable for all aspects of the work. TY: substantial contribution to the conception and design of the work; drafting the work and critical revisions; approval of final manuscript; accountable for all aspects of the work. VG: substantial contribution to the conception and design of the work; drafting the work and critical revisions; approval of final manuscript; accountable for all aspects of the work. BH: substantial contribution to the conception and design of the work; drafting the work and critical revisions; approval of final manuscript; accountable for all aspects of the work. CC-G: substantial contribution to the conception and design of the work; drafting the work and critical revisions; approval of final manuscript; accountable for all aspects of the work. CB: substantial contribution to the conception and design of the work; acquisition of data for the work; drafting the work and critical revisions; approval of final manuscript; accountable for all aspects of the work. FM: substantial contribution to the conception and design of the work; acquisition of data for the work; drafting the work and critical revisions; approval of final manuscript; accountable for all aspects of the work. IJ: substantial contribution to the conception and design of the work; acquisition of data for the work; drafting the work and critical revisions; approval of final manuscript; accountable for all aspects of the work. AD-O: substantial contribution to the conception and design of the work; acquisition of data for the work; drafting the work and critical revisions; approval of final manuscript; accountable for all aspects of the work. AC: substantial contribution to the conception and design of the work; acquisition of data for the work; drafting the work and critical revisions; approval of final manuscript; accountable for all aspects of the work.

**Funding** This research was funded by the NIHR CLAHRC Yorkshire and Humber - www.clahrc-yh.nihr.ac.uk (study funding number IS-CLA-0113-10020) and supported by the NIHR CLAHRC South West Peninsula and West Midlands CLAHRC.

**Disclaimer** The views and opinions expressed are those of the author(s), and not necessarily those of the NHS, the NIHR or the Department of Health and Social Care.

**Competing interests** None declared.

**Patient consent for publication** Not required.

**Ethics approval** This study was approved by the NRES Committee Yorkshire &and the Humber— Bradford Leeds on 10 October 2014 (14/YH/1120).

**Provenance and peer review** Not commissioned; externally peer reviewed.

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
