## [Reviewer comments · BMJ Open]

ARTICLE DETAILS

TITLE (PROVISIONAL)	The Community Ageing Research 75+ Study (CARE75+): an experimental ageing and frailty research cohort
AUTHORS	Heaven, Anne; Brown, Lesley; Young, John; Teale, Elizabeth; Hawkins, Rebecca; Splisbury, Karen; Mountain, Gail; Young, T; Goodwin, Victoria; Hanratty, Barbara; Chew-Graham, Carolyn; Brundle, Caroline; Mahmood, Farhat; Jacob, Ikhlaq; Daffu-O'Reilly, Amrit; Clegg, Andrew

VERSION 1 – REVIEW

REVIEWER	Merja Rantakokko University of Jyväskylä, Finland
REVIEW RETURNED	09-Oct-2018

GENERAL COMMENTS	This is an interesting protocol paper. Guidelines of BMJ Open require dates when data collection started, and this is missing from the paper. Please add information on timetable of the study. Ethical committee statement was given 4 years ago and related RCT registered 2016, thus it is a bit of concern whether this is an ongoing study (which it should be to be able to report it as a protocol article). I was also hoping to have clarification to recruitment protocol and sample size calculation. Plan is to recruit 1000 participants and " a minimum of 250 participants per year thereafter". Is the goal to recruit 250 at the baseline and 250 for next 3 years to reach in total 1000 participants, or do you plan to recruit first 1000 and then 250 for every year? For how long is the recruitment going to last? Add description of the current phase of the study. Probably a figure of study flow would be helpful?
---

REVIEWER	Emiel Hoogendijk Amsterdam UMC - location VU University Medical Center, Amsterdam, the Netherlands
REVIEW RETURNED	23-Oct-2018

GENERAL COMMENTS	This is an interesting research protocol of a very promising study. The protocol provides a comprehensive overview of the study. However, I have some comments for the authors to consider: - In addition to listing all assessments in text, an overview/summary of all measures could be provided in a figure or table.- The design of the study could be presented in a Figure (flowchart), so that the reader better understands the design and when new samples are recruited (250 each following year?).- It is a bit strange that a CARE75+ study has recently been
---

	published in Age and Ageing, without being mentioned in this manuscript (“Convergent validity of the electronic frailty index”).  - If the study already started (see previous comment), they authors may consider to present baseline data in this paper. If the study already started, please include dates (starting month, year). - The authors could diversify the cited literature, to include references outside of their own group. This provides a better context of their research. For example, two papers on the eFI have been published recently by other groups. Another example: some statements in this manuscript are only supported by opinion papers that the authors wrote themselves (reference no.6). - Using the original cut-offs of the Frailty phenotype (Fried et al.) may result in higher prevalence of frailty than usual, because the current study has an older population (75+) than the CHS population where Fried et al. validated their measure. The lowest quintile approach (for gait speed and grip strength) may be a better alternative. - The sample size: the focus of this study is on frailty. However, the study includes all people aged 75+, not only frail people. I do not see any projections on the number of people with frailty that will be included. Are these numbers sufficient? Especially when effect-modification and subgroup analyses are planned, the number of frail people needed may be higher. - The sample size calculation does not take into account that attrition rates may be higher when RCTs are performed among participants of this cohort study. - It is not clear what proportion of participants will be from ethnic minorities. If this proportion is substantial, a more systematic approach is needed. Only involving a person to translate (without proper translation of materials) will result in lower quality data. - If it is important to provide insight into the situation of ethnic minorities, these groups should be oversampled. Just including them, without oversampling, gives a representative sample for the overall older 75+ population, but does not result in representative samples from minority groups. This is very often the case in population-based cohort studies.
--	--

VERSION 1 – AUTHOR RESPONSE

Reviewer 1		
Please add information on timetable of the study. Ethical committee statement was given 4 years ago and related RCT registered 2016, thus it is a bit of concern whether this is an ongoing study (which it should be to be able to report it as a protocol article).	This is indeed an ongoing study. The first participant was recruited in January 2015. We have added this date to the manuscript.	P4 p1
Plan is to recruit 1000 participants and " a minimum of 250 participants per year thereafter". Is the goal to recruit 250 at the baseline and 250 for next 3 years to reach in total 1000 participants, or do you plan to recruit first 1000 and then 250 for every year? For how long is the recruitment going to last?	The plan is to recruit 1000 participants, and then 250 for every year. Recruitment is planned to last until September 2019, but we are awaiting a decision on funding extension to September 2024.	
Add description of the current phase of the study. Probably a figure of study flow would be helpful?	We have added a study flow diagram (figure 1). We are still at the recruitment stage, with the first participant expected to have final follow-up in January 2019.	Figure 1
Reviewer 2		
In addition to listing all assessments in text, an overview/ summary of all measures could be provided in a figure or table.	Thanks for this suggestion. A table has been added.	Table 1
The design of the study could be presented in a Figure (flowchart), so that the reader better understands the design and when new samples are recruited (250 each following year?).	A flowchart has been added.	Figure 1
It is a bit strange that a CARE75+ study has recently been published in Age and Ageing, without being mentioned in this manuscript ("Convergent validity of the electronic frailty index").	We have referenced the publication in the text.	References

If the study already started (see previous comment), they authors may consider to present baseline data in this paper. If the study already started, please include dates (starting month, year).	We plan to draft a separate cohort profile paper - this would be the usual way to present baseline data, rather than in the protocol paper.	
The authors could diversify the cited literature, to include references outside of their own group. This provides a better context of their research. For example, two papers on the eFI have been published recently by other groups. Another example: some statements in this manuscript are only supported by opinion papers that the authors wrote themselves (reference no. 6).	We have added reference to guideline documents that support statements.	References
Using the original cut-offs of the Frailty phenotype (Fried et al.) may result in higher prevalence of frailty than usual, because the current study has an older population (75+) than the CHS population where Fried et al. validated their measure. The lowest quintile approach (for gait speed and grip strength) may be a better alternative.	This is an important question. It is vital that the original cut-offs are used so that a standardised approach to frailty is taken. Otherwise, the definition of the condition will depend on the context in which it was assessed, which is inappropriate. The prevalence of frailty is anticipated to be higher in an over 75 population, so this is appropriate.	
The sample size: the focus of this study is on frailty. However, the study includes all people aged 75+, not only frail people. I do not see any projections on the number of people with frailty that will be included. Are these numbers sufficient? Especially when effect-modification and subgroup analyses are planned, the number of frail people needed may be higher.	Preliminary estimates indicate that around 80% of participants have either pre-frailty or frailty, so we are confident that number will be sufficient.	
The sample size calculation does not take into account that attrition rates may be higher when RCTs are performed among participants of this cohort study.	We agree, although the attrition rate for this novel approach is uncertain. We anticipate that attrition rates will indeed likely be higher for the nested RCTs but the impact on overall cohort attrition rates is an important question to investigate for the next phase of the work.	

It is not clear what proportion of participants will be from ethnic minorities. If this proportion is substantial, a more systematic approach is needed. Only involving a person to translate (without proper translation of materials) will result in lower quality data.	We have designed the cohort to be inclusive and, as far as possible, to represent the ethnic diversity of the older population. For example, around 15% of Bradford residents are from the south Asian (Pakistani) and a similar proportion of CARE75+ participants in Bradford have been recruited.	
If it is important to provide insight into the situation of ethnic minorities, these groups should be oversampled. Just including them, without oversampling, gives a representative sample for the overall older 75+ population, but does not result in representative samples from minority groups. This is very often the case in population-based cohort studies.	Many thanks. We agree that this is an important consideration. We are considering oversampling of particular minority ethnic groups as a component of the next phase of CARE75+, if ongoing funding is secured.	

VERSION 2 – REVIEW

REVIEWER	Emiel Hoogendijk VU University Medical Center, Amsterdam, the Netherlands
REVIEW RETURNED	28-Nov-2018
GENERAL COMMENTS	The authors have addressed all comments. They have added a table and a flowchart, however, they decided not to incorporate the remaining comments. That is their choice, and partly understandable. However, as a result, they missed the opportunity to provide details on specific elements of the study (such as sampling of minority groups).